# CO_2_ LASER versus Blade Scalpel Surgery in the Management of Nasopharyngeal Masses in Dogs

**DOI:** 10.3390/ani14121733

**Published:** 2024-06-08

**Authors:** L. Miguel Carreira, Graça Alexandre-Pires, Pedro Azevedo

**Affiliations:** 1Anjos of Assis Veterinary Medicine Centre—CMVAA, 2830-077 Barreiro, Portugal; pedro.almeida.azevedo@gmail.com; 2Faculty of Veterinary Medicine, University of Lisbon (FMV/ULisboa), 1300-477 Lisboa, Portugal; gpires@fmv.ulisboa.pt; 3Interdisciplinary Centre for Research in Animal Health (CIISA), University of Lisbon (FMV/ULisboa), 1300-477 Lisboa, Portugal; 4Associate Laboratory for Animal and Veterinary Sciences (AL4AnimalS), 1300-477 Lisbon, Portugal; 5Faculty of American Laser Study Club—ALSC, Altamonte Springs, FL 32714, USA

**Keywords:** dog, surgery, nasopharyngeal oncology, scalpel, CO_2_ laser

## Abstract

**Simple Summary:**

In this study, we compared the outcomes of surgery for nasopharyngeal tumors in dogs using a conventional scalpel blade and a CO_2_ surgical laser system. We aimed to evaluate surgical time, bleeding level, patient pain, healing time, scar tissue formation, tumor relapse, and complications. We conducted a clinical prospective study involving 12 inpatient dogs, with 6 undergoing surgeries with a scalpel blade and 6 with a CO_2_ laser. We found that surgery using the CO_2_ laser resulted in shorter surgical and healing times, lower pain levels, less scar tissue formation, and a lower rate of tumor relapse compared to scalpel blade surgery. Our findings suggest that using the CO_2_ laser for nasopharyngeal tumor surgery may offer advantages in terms of precision, reduced discomfort, and fewer complications.

**Abstract:**

We aimed to compare surgical time, bleeding level, patient pain level, healing period, scar tissue, relapse of the initial process and complications in patients with nasopharyngeal oncological masses undergoing surgery using a scalpel blade versus a CO_2_ surgical laser. This is a clinical prospective study comprising surgical work in the nasopharynx area. A sample of 12 inpatients dogs (N = 12) of both genders underwent a surgical excision of nasopharyngeal masses with a scalpel blade (GS *n* = 6) and CO_2_ surgical laser (GL *n* = 6). An Aesculigth CO_2_ surgical laser-Vetscalpel^®^ model with a superpulse mode, 12 W of power, and a multi-focus pen was used. Statistically significant differences were registered for a *p*-value of < 0.05. Variations were noted between both groups (GS and GL) concerning surgery time (*p* = 0.038), first meal time (*p* = 0.013), pain level (*p =* 0.003), and healing time (*p =* 0.014), with the GL group presenting lower values. GL exhibited only one relapse case, with the elapsed time being more than double that of the GS group. Surgical and healing times were shorter in the GL group, and pain levels were lower, with the GL group also demonstrating less scar tissue than the GS group, along with a lower relapse rate. Nasopharynx surgical exposure with precision via the soft palate using the CO_2_ laser has facilitated successful treatment of regional masses without discomfort and complications, compared to conventional blade scalpel procedures.

## 1. Introduction

The pharynx, a crucial anatomical structure in both humans and animals, comprises three distinct regions: the naso-, oro-, and laryngopharynx. Each of these regions plays a vital role in various physiological processes, including respiration, swallowing, and vocalization. The nasopharynx, positioned between the choanae, soft palate, base of the skull, and larynx, serves as a conduit for airflow during breathing and is involved in the production of mucus to humidify and protect the respiratory tract. In veterinary medicine, canine primary pharyngeal cavity tumors represent a notable pathology, constituting approximately 2–4% of neoplastic diseases in dogs. These tumors often manifest as expansive masses characterized by a high degree of local invasiveness and an infiltrative growth pattern [1,2,3]. In dogs, the oropharynx is the most frequently affected site, followed by the laryngopharynx, with the nasopharynx being the rarest location for tumor development [1,2,4,5,6,7]. Most nasopharyngeal tumors in dogs arise in specific anatomical regions, such as the fossa of Rosenmuller, and tend to exhibit deep-seated spread growth patterns [8,9]. Notably, gender dimorphism has been observed in the incidence of pharyngeal tumors, with males having a 2.4 times higher risk of developing such tumors compared to females [10]. This gender disparity may stem from hormonal or genetic factors influencing tumor initiation and progression. Oncological surgery in the pharyngeal region poses significant challenges due to the anatomical complexities and proximity to critical structures, such as major blood vessels, nerves, and the respiratory and digestive tracts. Achieving complete tumor removal while preserving vital structures is often very difficult to perform, and the risk of intraoperative complications remains a concern [4,8,9,10,11,12]. Recent advances in surgical technology have revolutionized the management of pharyngeal tumors in both human and veterinary medicine. Among these innovations, the carbon dioxide (CO_2_) laser has emerged as a powerful tool in maxillofacial/head and neck oncological surgeries [13,14,15,16,17,18,19,20]. The CO_2_ laser falls under the WYSIWYG class (“What You See Is What You Get”), boasting exceptional precision for surgical interventions. Selective absorption by water molecules in tissues enables precise tissue ablation with minimal thermal damage to the surrounding structures, making it particularly well suited for delicate procedures in the oral cavity and pharynx, such as for transoral mass resections [8,9,10,11,12,13,14,15,21,22,23,24,25]. The aim of the present study was to compare the outcomes of patients with nasopharyngeal oncological masses undergoing surgical resection using conventional scalpel blades versus CO_2_ surgical laser technology. Parameters assessed included surgical duration, intraoperative bleeding volume, postoperative pain levels, healing time, formation of scar tissue, tumor recurrence, and incidence of intra- or postoperative complications. By evaluating these parameters, we aimed to provide insights into the efficacy and safety of CO_2_ laser-assisted surgery in the management of nasopharyngeal tumors, potentially informing future treatment strategies and improving patient outcomes.

## 2. Materials and Methods

This study started on 1 January 2022 and concluded on 31 January 2024. It involved a sample of 12 dogs (N = 12) of both genders, breeds, and ages, all diagnosed with malignant nasopharyngeal processes initially characterized by cytology. These dogs were client-owned patients. Prior to their inclusion in this study, their caregivers signed an informed consent form, allowing participation in this study. Prior to surgery, each patient underwent peripheral blood sample collection from the cephalic vein after topical application of bupivacaine gel to perform a hemogram as well as liver and kidney function analysis. Additionally, all patients underwent fine needle aspiration of regional lymph nodes, and thoracic radiographs were obtained using right and left lateral projections to assess signs of lung metastases (tumor staging). A head CT of all patients was obtained to assess the extent of the tumor, regional metastasis, and bone osteolytic lesions in order to prepare for surgery. Thoracic CT images were obtained for only 4 of the 12 patients.

The dogs were divided into two groups: one group underwent surgical excision with a scalpel blade (GS group), while the other group underwent surgery using the CO_2_ surgical laser (GL group), reflecting the preference of the caregivers for the surgical technique. To ensure consistency, all patients received the same therapeutic protocol with fluid therapy NaCl 0.9% (2 mL/kg/hr/IV), Cefazolin (20 mg/kg/IV), Buprenorphine (10 μg/kg/IM), Acepromazine (0.02 mg/Kg/IV), and Carprofen (4 mg/kg/SC). For anesthesia induction, Propofol (4 mL/kg) was used, and its maintenance was made with isoflurane. Surgeries were performed by the same surgeon, along with the scar tissue evaluation of the intervened region, to minimize bias. For the scalpel blade technique, electrocoagulation was used to ensure hemostasis. For the CO_2_ surgical laser procedures, an Aesculight CO_2_ laser model Vetscalpel^®^ was utilized, set to a continuous wave in superpulse mode with an emission of 12 watts of power. A multi-focus pen with a 0.4 mm tip for dissection and a 1.2 mm tip for vaporization of the target area was employed. The surgical procedure started with a midline soft palate incision overlying the tumor, followed by careful laser resection and blunt dissection from surrounding structures. Tumor excision was possible to perform as a bloc (*en bloc*). The excised nasopharyngeal masses were then sent for histopathological characterization. To assess the patient’s postoperative pain level, we used the Melbourne Pain Scale (MPS). The Melbourne Pain Scale is a standardized and validated multi-parameter tool to assess pain levels in dogs. It evaluates several indicators, such as behavior, physiology, and context, to determine the intensity of pain experienced by the animal. Evaluation of the local scar tissue was performed by the surgeon in all patients at day 10 post-surgery by opening the patient’s mouth to visualize the scar’s appearance and by touching it with the index finger to assess its softness or roughness. Sedation was not necessary for this procedure in any patient. All patients underwent a follow-up over a period of 6 months (180 days) during the study. Statistical analyses were conducted using R software (version 3.01). Normality of data distribution was assessed using the Shapiro–Wilk test, and comparisons between GS and GL parameters were made using one-way ANOVA. Statistically significant differences were registered for a *p*-value of < 0.05, indicating variations between the two surgical techniques in terms of pain management, surgical outcomes, and postoperative recovery.

## 3. Results

Twelve patients were enrolled in this study. They were divided equally between the GS (scalpel blade) and GL (CO_2_ laser) groups, and their characterization is presented at Table 1. No significant differences were observed between the two groups regarding the following parameters: age (*p =* 0.455), body weight (*p =* 0.725), and tumor mass dimensions. Statistically significant differences were registered in several parameters between the GS and GL groups. These included surgical time (*p =* 0.038), patient first meal time (*p =* 0.013), patient pain level (*p =* 0.003), and tissue healing time (*p =* 0.014). Local scar tissue was evaluated visually and via palpation of the region, with the GL group presenting a shorter and softer scar than the GS group. Regarding MPS scores, the GL group presented an MPS = 0 compared with an MPS score of 1.16 ± 0.75 of the GS group at the end of the first 24 h post-surgery (Table 1). For GL patients, the average time for their first meal was 8 h after the end of surgery, without showing any discomfort when eating. In contrast, in GS patients, the mean time registered for their first meal was 12 h after the end of surgery (Figure 1). This difference was statistically significant (*p* = 0.026). Patients in the GL group exhibited shorter surgical times, earlier initiation of oral intake, lower pain levels, and faster tissue healing compared to those in the GS group. Visual and palpation assessments of scar tissue indicated that patients in the GL group had shorter and softer scars compared to those in the GS group. The histopathological diagnosis of the excised nasopharyngeal masses was similar in both groups, with the following results: squamous cell carcinoma (4 cases in total: 3 in the GS group and 1 in the GL group) was the most prevalent, followed by adenocarcinoma (3 cases in total: 2 in the GS group and 1 in the GL), fibrosarcoma (2 cases in total: 1 in each group), undifferentiated sarcoma (2 cases in total: only in the GL group), and lymphoma (1 case in total: only in the GL group). Additionally, melanoma was concurrently detected in one patient in the GL group, although it was not associated with the nasopharynx but rather with the lower lip of the patient. The number of relapse cases were more frequent in the GS group (4 cases) compared to the GL group (1 case). Furthermore, the average time for the diagnosis of the first clinical signs of relapse was significantly longer in the GL group (138 days) compared to the GS group (52 days), indicating the potential efficacy of CO_2_ laser surgery in achieving longer-term outcomes (Table 2).

## 4. Discussion

The pharynx is the common interface between the digestive and respiratory tracts. Due to its anatomical location, nasopharyngeal diseases are often difficult to visualize during a physical examination in the early stages. Patients may exhibit nonspecific clinical signs and symptoms, mimicking common upper respiratory tract diseases, thereby delaying the diagnosis. According to the study results, stertor, oral breathing, nasal discharge, and anorexia were the most significant clinical signs presented by the patients, which is in accordance with the studies of Hunt [2,4] and Rodriguez et al. [26], who concluded that patients may present with both respiratory and digestive clinical signs. Nasopharyngeal masses may clinically manifest only after they become infected, thereby becoming more swollen and larger, which induces the onset of clinical signs. The use of advanced imaging techniques, such as computed tomography (CT) and magnetic resonance imaging (MRI), is the gold standard for the investigation of tumors in this anatomical region. These techniques allow for the evaluation of the location and extension of the tumor as well as bone involvement in the affected area [27,28]. All the patients in this study underwent head CT to characterize the nasopharyngeal mass and to prepare for their surgical intervention. Additionally, all patients had thoracic radiographs to rule out the possibility of pulmonary metastases, which were not diagnosed. These results are in line with the study by Bolwska et al. [27], which concluded that metastases from nasal-region tumors to the lungs are uncommon. It is recognized that the gold standard of care for nasal and nasopharyngeal tumors typically involves radiotherapy as the primary treatment modality [29,30,31,32]. Radiotherapy utilizes high-energy X-rays and encompasses three main types: intensity-modulated radiation therapy, stereotactic radiation therapy, and palliative intent radiation therapy. Despite its potential, radiotherapy can substantially enhance the quality of life and tumor control, leading to approximately 90% of affected pets showing a clinical improvement post-treatment due to tumor volume reduction. Notably, certain tumor types, such as sarcomas, may exhibit slower and less-pronounced shrinkage. However, accessibility to this treatment is significantly limited for the majority of patients due to challenges in accessing specialized facilities and the associated high costs, which are often prohibitive from an economic perspective for many owners. Consequently, surgical resection remains the most common treatment approach, which is sometimes associated with chemotherapy or radiotherapy.

The present study compared the outcomes of patients with nasopharyngeal oncological masses undergoing surgical resection using conventional scalpel blades versus CO_2_ surgical laser technology. After discussing the benefits of the CO_2_ laser surgery technique with caregivers, such as reduced bleeding, lower patient anesthesia consumption, shorter surgery time, less postoperative pain, faster healing, and improved local scarring compared to the conventional scalpel blade technique [17], caregivers were able to choose which technique to use. Those who chose the conventional scalpel blade technique justified their decision based only on the economic difference between the two techniques.

Although no intra- or postoperative complications were seen in either group, it was possible to conclude that the comparison between conventional surgery with a blade scalpel and CO_2_ laser surgery for nasopharyngeal oncology masses revealed notable advantages associated with the latter technique. Specifically, the CO_2_ laser approach demonstrated shorter surgical and healing times when compared to traditional scalpel-based surgery. This expedited surgical process can be attributed to the CO_2_ laser’s unique photothermal properties, which facilitate efficient tissue incisional procedures, vaporization of the tissue, and hemostasis within the oral cavity’s soft tissues.

The use of a CO_2_ laser in the nasopharyngeal region was found to be associated with significantly reduced intraoperative bleeding compared to conventional scalpel-based surgery [23,26,27,28,29,30,31,32,33,34,35,36,37,38,39]. With the precision of the CO_2_ laser, vessels with a diameter smaller than 0.5 mm are coagulated, effectively controlling intraoperative bleeding. As a result, a bloodless field with an excellent view of the region is provided, allowing the surgery to be performed more precisely and accurately because of the increased visibility on the surgical region’s structures. Hemostasis with the CO_2_ laser is achieved by the contraction of the collagen from the vascular wall of vessels, leading to a reduction in the vessel diameter and therefore controlling the bleeding levels [36,38,39,40,41].

The Melbourne Pain Scale (MPS) has been widely utilized in veterinary medicine and research, aiding in the development of improved pain management strategies and enhancing the outcomes for canine patients undergoing diverse surgical and medical procedures. In our study, we employed the MPS, facilitating a straightforward comparison of pain levels between the GS and GL groups. Our findings revealed superior pain control in the GL laser group during the initial 24 h postoperative period. The CO_2_ laser beam promotes the carbonization of sensory nerve endings, resulting in an immediate neural transmission blockage. As a result, the intra- and postoperative patient pain levels were lower [8,23,42].

With the CO_2_ laser, the dissection followed the approach of “*en bloc*” removal of tumor tissue, rather than an anatomically based dissection with the scalpel. As a result, more normal tissue can be preserved in patients, enhancing the functional preservation of the surgical region [8,17,21,23,38,43]. By inducing the instantaneous vaporization of the cell structure with little or no release of inflammation mediators, the CO_2_ laser promotes minimal damage to tissues; therefore, it is associated with a lower acute inflammatory reaction compared to conventional surgery. At the same time, the denaturation of matrix proteins (such as collagen) occurs, forming a layer on the cell surface that likely acts as an impermeable shield, thereby decreasing the total plasma protein loss. This results in a lower postoperative edema level compared to the blade scalpel technique [23,38,44,45]. Additionally, by decreasing postoperative swelling, the range of surgeries that surgeons can perform safely without the fear of airway compromise increases. This is a very important factor in all breeds, but especially in brachycephalic breeds [6,46].

Regarding the healing process, in terms of scar tissue formation, CO_2_ laser surgery resulted in significantly less-noticeable scarring compared to conventional scalpel-based surgery. It is known that the energy emitted by the surgical CO_2_ laser affects the collagen architecture and intracellular water content in tissues, inducing a skin contraction phenomenon. This contraction offers surgical advantages, notably in reducing tissue invasion, resulting in softer and more aesthetically pleasing scars [47].

Although the quantification of scar tissue differences between both groups using a digital caliper was desirable [23,30,38,44], this task was not possible to carry out due to logistical constraints because it required the patient to be anesthetized, which was not justified for this purpose. Nasopharyngeal tumors in dogs present surgical challenges due to their location and due to the potential involvement of critical structures, such as the nasal passages, pharynx, and surrounding tissues. Relapse cases of these tumors in dogs occur when the tumor reappears after initial treatment or remission. These relapses can result from various factors, including incomplete tumor removal during surgery, tumor size, histological type, extent of local invasion, and resistance to treatment modalities (such as chemotherapy or radiation therapy), among other factors. Treatment options for relapsed nasopharyngeal tumors in dogs may include additional surgery, chemotherapy, radiation therapy, or a combination of these modalities. The choice of treatment depends on the individual patient’s condition, the preferences of the caregivers, and the availability of resources for accessing various treatment techniques [46,47,48,49,50,51]. By comparing relapse cases, CO_2_ laser surgery exhibited a lower incidence, with only one case compared to four cases in the conventional surgery group. The extended elapsed time until the first clinical signs in the CO_2_ laser group was more than double that in blade scalpel group, indicating its potential efficacy in long-term outcomes.

The literature on veterinary medicine regarding surgical work in this particular nasopharynx region is very uncommon, underscoring the novelty and significant contribution of this study’s findings to veterinary medicine. The absence of comparable previous works underscores the novelty and importance of these results, highlighting the need for further research in this area to validate and expand upon these observations.

## 5. Conclusions

The CO_2_ laser’s ability to provide precise nasopharyngeal surgical exposure through the soft palate made an accurate treatment of masses in this region possible, overcoming the historical challenges associated with accessing and visualizing this anatomical region. This breakthrough has not only simplified the procedure but has also significantly reduced patient discomfort and the likelihood of complications. With the present study, we have highlighted numerous advantages for both patients and surgeons when using the CO_2_ laser for resecting nasopharyngeal oncology masses compared to traditional blade scalpel procedures. The precision and control offered by the CO_2_ laser enable surgeons to navigate the intricacies of the nasopharynx with unparalleled accuracy, resulting in improved surgical outcomes and a reduced risk of complications. Moreover, the minimally invasive nature of CO_2_ laser surgery translates to faster recovery times and less postoperative discomfort for patients. By minimizing tissue damage and preserving surrounding structures, the CO_2_ laser facilitates a more gentle and effective approach to mass resection in the nasopharynx. Overall, our findings underscore the transformative impact of CO_2_ laser technology in the field of veterinary surgery, particularly in the treatment of nasopharyngeal masses. By offering a safer, more efficient, and less-invasive alternative to conventional methods, the CO_2_ laser represents a promising advancement in veterinary surgical practice for treating nasopharyngeal masses.

## Figures and Tables

**Figure 1 animals-14-01733-f001:**
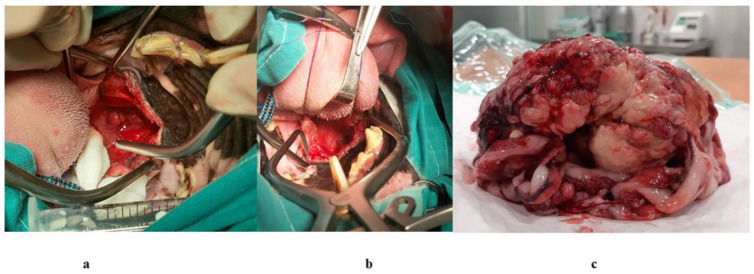
Dissecting the nasopharyngeal mass with the CO_2_ laser (**a**); the final aspect of the interviewed anatomical area (**b**); all the mass removed as a block (**c**).

**Table 1 animals-14-01733-t001:** Sample characterization parameters evaluated in this study, comparing the type of surgery-scalpel versus CO_2_ laser.

Parameter				N	Type of Surgery Technique			
					Scalpel (*n* = 6)		CO_2_ Laser (*n* = 6)	
					Mean ± SD		Mean ± SD	
Gender				12	Female	*n* = 3	Male	*n* = 9
Age (years)				12	11.9 ± 2.65		10.6 ± 3.12	
Weight (kg)				12	19.29 ± 4.81		20.21± 4.05	
Surgical time (min)				12	84 ± 14.7		64.6 ± 13.4	
Mass dimension (mm)	Total sample	Length	5.30 ± 2.19	12	3.71 ± 1.13		7.53 ± 1.08	
		Width	4.02 ± 1.44	12	3.25 ± 1.26		5.11 ± 0.87	
		Height	4.24 ± 2.06	12	3.26 ± 1.77		5.61 ± 160	
First meal time (hours)				12	11.6 ± 3.2		7.3 ± 1.5	
Pain level (24 h later)				12	1.16 ± 0.75		0	
Time to scar finally healing (days)				12	11.6 ± 2.06		8.5 ± 1.64	
Relapse				12	Yes (*n*= 4)		Yes (*n* = 1)	
First clinical signs associated with relapse				12	52.16 ± 47.1		138	
Patients’ clinical signs				12	**Clinical signs**		**%**	
					stertor		91.7	
					oral breathing		83.3	
					nasal discharge		66.7	
					anorexia		66.7	
					cough		50	
					dysphagia		41.7	
					sneezing		33.3	
					ptyalism		33.3	
					vomiting		25	
**Type of tumor**				** *n* **	**Scalpel (*n*)**		**CO_2_ laser (*n*)**	
Squamous cell carcinoma				4	3		1	
Adenocarcinoma				3	2		1	
Fibrosarcoma				2	1		1	
Undifferentiated sarcoma				2	-		2	
Lymphoma				1	-		1	

**Table 2 animals-14-01733-t002:** One-way ANOVA test for comparison between patients subjected to scalpel blade versus CO_2_ laser surgery for nasopharynx oncological mass surgery.

Parameter		One-Way ANOVA Test	
		F-Ratio	*p*-Value
Age (years)	Blade scalpel vs. CO_2_ laser	0.603	0.455
Body weight (kg)	Blade scalpel vs. CO_2_ laser	0.129	0.725
Surgical time (min)	Blade scalpel vs. CO_2_ laser	5.636	0.038 *
First meal time (h)	Blade scalpel vs. CO_2_ laser	8.989	0.013 *
Pain level (24 h later)	Blade scalpel vs. CO_2_ laser	14.411	0.003 *
Time to scar finally healing (days)	Blade scalpel vs. CO_2_ laser	8.636	0.014 *

* Statistically significant differences for *p*-value > 0.05.

## Data Availability

Data are contained within the article.

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
