# Peer review of "CO2 LASER versus Blade Scalpel Surgery in the Management of Nasopharyngeal Masses in Dogs"

_animals, 2024, doi:10.3390/ani14121733_

Round 1

Reviewer 1 Report

Comments and Suggestions for Authors

 Thank you for this interesting paper however I would recommand few changes to improve the quality of this article: 

Why is the term "LASER" capitalized? I believe it's just a regular noun, not a proper noun, so it should be written “laser” like other common nouns throughout the text.

I assume that the first sentence is a mistake: “The introduction should briefly place the study in a broad context and highlight 44 whyThe pharynx, a crucial anatomical structure in both humans and animals, comprises 45 three distinct regions: the naso-, oro-, and laryngopharynx.”

During the scalpel procedure, was electrocoagulation or any other method used to ensure hemostasis?

How was the scar evaluated if it was located in an area inaccessible for examination? Are there any photos of scar after scalpel and scar after laser? Was the scar assessment conducted during premedication? I think the result about scar formation should be removed from the result section and can be a part of the discussion but in my opinion the  subjective evaluation of the scar is not a valuable scientific data.

Were antibiotics administered to the patients?

Was nasal packing necessary for the patients?

Comments on the Quality of English Language

 Why is the term "LASER" capitalized? I believe it's just a regular noun, not a proper noun, so it should be written “laser” like other common nouns throughout the text.

Change "teh" to "the" in line 153.

Please remove Bold in letter T line 111. 

Author Response

Dear Reviewer,

I hope this message finds you well. Thank you for taking the time to review our manuscript titled "CO2 LASER VERSUS BLADE SCALPEL SURGERY IN THE MANAGEMENT OF NASOPHARYNGEAL MASSES IN DOGS," which we have submitted for publication in the Animals. We appreciate your valuable feedback, and we have carefully considered all of your comments. In response, we have made revisions to the manuscript to address each point raised. Below, you will find our detailed responses to your comments, organized to correspond with the changes made in the original version of the manuscript.

Thank you once again for your time and thoughtful input.

Best regards,

L.Miguel Carreira

Why is the term "LASER" capitalized? I believe it's just a regular noun, not a proper noun, so it should be written “laser” like other common nouns throughout the text.

LASER is an acronym that stands for "Light Amplification by Stimulated Emission of Radiation." Each letter in LASER corresponds to the first letter of each word in the phrase, which describes the fundamental process by which lasers produce light. That is why we initially used it in capitalized form in the manuscript. However, according to your suggestion, we have decided to substitute all instances of LASER in capitalized form with "laser" in regular letters.

I assume that the first sentence is a mistake: “The introduction should briefly place the study in a broad context and highlight 44 whyThe pharynx, a crucial anatomical structure in both humans and animals, comprises 45 three distinct regions: the naso-, oro-, and laryngopharynx.”

The reviewer is totally right in stating that there is an error in the phrase 'The introduction should briefly place the study in a broad context and highlight why the pharynx, a crucial anatomical structure in both humans and animals, comprises three distinct regions: the naso-, oro-, and laryngopharynx.' It has been replaced with the following: 'The pharynx, a crucial anatomical structure in both humans and animals, comprises three distinct regions: the naso-, oro-, and laryngopharynx.'

During the scalpel procedure, was electrocoagulation or any other method used to ensure hemostasis?

In response to the question, during the scalpel procedure, we used compresses and electrocoagulation to ensure hemostasis. We decided to add the following sentence “For the scalpel blade technique, electrocoagulation was used to ensure hemostasis.”

How was the scar evaluated if it was located in an area inaccessible for examination? Are there any photos of scar after scalpel and scar after laser? Was the scar assessment conducted during premedication? I think the result about scar formation should be removed from the result section and can be a part of the discussion but in my opinion the  subjective evaluation of the scar is not a valuable scientific data.

Although we agree with the reviewer that scar tissue evaluation can be subjective, we believe it is important to maintain the reference on the subject since it shows that there are differences between both techniques, with a better result for the laser patients. Also, the parameter in question was always assessed by the same investigator - the surgeon, thus reducing the bias associated with its classification. Local scar assessment was performed on all patients by visual inspection and local palpation. The procedure was easily carried out by opening the patient's mouth and directly visualizing and palpating the area, as the incision was made at the level of the soft palate as described in the Materials and Methods section. Thus, we decided to add the following sentence: “Evaluation of the local scar tissue was performed by the surgeon in all patients at day 10 post-surgery, by opening the patient mouth to visualize the scar's appearance and touching it with the index finger to assess its softness or roughness. Sedation was not necessary for this procedure in any patient.”.

Were antibiotics administered to the patients?

Yes, we administered antibiotics to all patients. In response to the question, we decided to add the following sentence to the Materials & Methods section: “All patients received the same therapeutic protocol with fluid therapy NaCl 0.9% (2 ml/kg/hr), Cefazolin (20 mg/kg), Buprenorphine (10 μg/kg), Acepromazine (0.02 mg/kg), and Carprofen (6 mg/kg). For anesthesia induction, Propofol (4 ml/kg) was used, and maintenance was done with isoflurane”.

Was nasal packing necessary for the patients?

No nasal packing was necessary for any patient in either of the groups

Change "teh" to "the" in line 153.

Corrected

Please remove Bold in letter T line 111. 

Corrected

Reviewer 2 Report

Comments and Suggestions for Authors

The authors are thanked for the presentation of their work.

Although the paper is interesting, there are certain points that make it unsuitable for publication, such as the following:- The study has neither passed nor been accepted by an ethics committee.

- The treatment of choice described in the literature and based on evidence for nasopharyngeal tumours such as lymphoma, whose treatment of choice is radiochemotherapy and not surgery, has not been taken into account.

- The text does not indicate the results of the cytology of the mass and lymph nodes prior to surgery, nor whether alterations were observed in the chest radiographic study. 

- It does not indicate whether or not a cranial CT scan was performed to assess the extent of the tumour, regional metastasis and osteolytic lesion.

- It does not indicate the weight or age range of the patients, nor how many were female or male.

Author Response

Dear Reviewer,

I hope this message finds you well. Thank you for taking the time to review our manuscript titled "CO2 LASER VERSUS BLADE SCALPEL SURGERY IN THE MANAGEMENT OF NASOPHARYNGEAL MASSES IN DOGS," which we have submitted for publication in the Animals. We appreciate your valuable feedback, and we have carefully considered all of your comments. In response, we have made revisions to the manuscript to address each point raised. Below, you will find our detailed responses to your comments, organized to correspond with the changes made in the original version of the manuscript.

Thank you once again for your time and thoughtful input.

Best regards,

Although the paper is interesting, there are certain points that make it unsuitable for publication, such as the following:- The study has neither passed nor been accepted by an ethics committee.

The study was approved by the ethics committee with the reference number 015/2022, as described in the item Institutional Review Board Statement:”The study was conducted in accordance with the Declaration of Helsinki, and approved by the Institutional Review Board (or Ethics Committee) of Faculty of Veterinary Medicine - University of Lisbon (protocol code 015/2022), altought the study did not required ethical approval because it was not an experimental study.”

The treatment of choice described in the literature and based on evidence for nasopharyngeal tumours such as lymphoma, whose treatment of choice is radiochemotherapy and not surgery, has not been taken into account.

Radiation therapy is a technique that is not available in the majority of large hospitals and veterinary medical centers, not only due to the difficulty in constructing and maintaining facilities for this purpose but also because it is a type of treatment that is almost prohibitive from an economic standpoint for most tutours. For this reason, it was not the topic addressed in the article. Nonetheless, we have decided to add the following paragraph to the discussion section: “It is recognized that the gold standard of care for nasal and nasopharyngeal tumors typically involves radiotherapy as the primary treatment modality [6,12,17]. Radiotherapy utilizes high-energy X-rays and encompasses three main types: intensity-modulated radiation therapy, stereotactic radiation therapy, and palliative-intent radiation therapy. Despite its potential, radiotherapy can substantially enhance quality of life and tumor control, leading to approximately 90% of affected pets showing clinical improvement post-treatment due to tumor volume reduction. Notably, certain tumor types, such as sarcomas, may exhibit slower and less pronounced shrinkage. However, accessibility to this treatment is significantly limited for the majority of patients due to challenges in accessing specialized facilities and the associated high costs, which are often prohibitive from an economic perspective for many tutors. Consequently, surgical resection remains the most common treatment approach, sometimes associated with chemotherapy or radiotherapy. The present study compared the outcomes of patients with nasopharyngeal oncological masses undergoing surgical resection using conventional scalpel blades versus CO2 surgical laser technology.”

- The text does not indicate the results of the cytology of the mass and lymph nodes prior to surgery, nor whether alterations were observed in the chest radiographic study. 

According to the reviewer suggestion we added the following sentence to Material & Methods: “Prior to surgery, each patient underwent peripheral blood sample collection from the cephalic vein after topical application of bupivacaine gel to perform hemogram and liver and kidney function analysis. Additionally, chest radiographs were obtained using right and left lateral projections for all patients to assess signs of lung metastases.”

- It does not indicate whether or not a cranial CT scan was performed to assess the extent of the tumour, regional metastasis and osteolytic lesion.

According to the reviewer suggestion we added the following sentence to Material & Methods: “ All patients were subjected to cranial CT scans to assess the extent of the tumor, regional metastasis, and osteolytic lesions, in order to prepare for surgery. Thoracic CT images were obtained for only 4 of the 12 patients.”

- It does not indicate the weight or age range of the patients, nor how many were female or male.

Table nº1 already presented the age and weight parameters of the sample. The gender was not referred but it was already added to the table.

Reviewer 3 Report

Comments and Suggestions for Authors

Review

CO2 LASER Versus Blade Scalpel Surgery in the Management 2 of Nasopharyngeal Masses in Dogs

The reviewer thanks the authors for this submission.

It is nice to see a comparison of conventional surgery and CO2 laser in a case series. The reviewer acknowledges that these tumors are not very common and therefore numbers in the study are low.

The overall study is set up like a prospective study. It is not fully clear if this is the case. There is no beginning or end date to the study.

Additionally, many items of evaluation are missing, mainly a more detailed clinical picture of the patients included in the study. A few more comments below:

Line 44  Please remove - The introduction should briefly place the study in a broad context and highlight why.

Line 56 - 60 strike similar to humans … -  – concentrate on dogs – can discuss people in discussion but concentrate on the species at hand with the intro. Intro should be to the point and precise.

Line 60-65 – mixing dogs and people in this again as references are listed for both. Please remove and only report on dogs.

Line 86 – what does this sentence mean? Were they so sick they had been admitted to the clinic prior to decision of surgery or are the authors meaning to say these dogs were client owned patients. Please clarify.

Line 86 – is tutor equivalent to dog owner?

Line 108 please put a space after <

Table 1 – the Mean should be a number < 50 mm and > 50 mm is not a mean. Please provide means with SD for this. 

The reviewer is missing a lot more results.

Full Signalment of patients is missing.

To introduce a new technique, imaging of the tumors with pictures of CTs and measurements would be very helpful.

Also mass dimensions with just one dimension present is not sufficient in advanced oncology.

Furthermore information regarding blood loss would be great to compare the two techniques. Additionally since the owners are allowed to choose the procedure they like done rather than randomly being assigned to a group more understanding on counseling and how the decisions were derived is important.

Please provide the anesthesia and analgesic protocol for the reviewers.

How was healing time evaluated? Were patients reexamined under sedation to examine the incision site? in line 188

Line 180 Lower edema level was assessed how? If patients were not thoroughly examined post surgery this is hard to compare and an explanation needs to be given that is more detailed than this.

Please provide information about follow up treatment for each case. Since there are only 6 cases a table with each case can be provided with vital information to better follow the cases. 

Author Response

Dear Reviewer,

I hope this message finds you well. Thank you for taking the time to review our manuscript titled "CO2 LASER VERSUS BLADE SCALPEL SURGERY IN THE MANAGEMENT OF NASOPHARYNGEAL MASSES IN DOGS," which we have submitted for publication in the Animals. We appreciate your valuable feedback, and we have carefully considered all of your comments. In response, we have made revisions to the manuscript to address each point raised. Below, you will find our detailed responses to your comments, organized to correspond with the changes made in the original version of the manuscript.

Thank you once again for your time and thoughtful input.

Best regards,

The overall study is set up like a prospective study. It is not fully clear if this is the case. There is no beginning or end date to the study. Additionally, many items of evaluation are missing, mainly a more detailed clinical picture of the patients included in the study.

Thank you for your insightful comments. In response, we have revised the manuscript to address these concerns, and we have enriched the manuscript with more detailed clinical profiles of the patients.

Line 44  Please remove - The introduction should briefly place the study in a broad context and highlight why.

The reviewer is totally right in stating that there is an error in the phrase. It was corrected

Line 56 - 60 strike similar to humans … -  – concentrate on dogs – can discuss people in discussion but concentrate on the species at hand with the intro. Intro should be to the point and precise.

According to the reviewer suggestion, the sentence “Similar to humans, most nasopharyngeal tumors in dogs arise in specific anatomical regions, such as the fossa of Rosenmuller, and tend to exhibit deep-seated spread growth patterns” was replaced by the following” Most nasopharyngeal tumors in dogs arise in specific anatomical regions, such as the fossa of Rosenmuller, and tend to exhibit deep-seated spread growth patterns”

Line 86 – what does this sentence mean? Were they so sick they had been admitted to the clinic prior to decision of surgery or are the authors meaning to say these dogs were client owned patients. Please clarify.

All the patients were client-owned. Some were internal patients, and others were referred by colleagues to our facilities for surgery. We decided to substitute the sentence with: 'These dogs were client-owned patients.

Line 86 – is tutor equivalent to dog owner?

Yes, 'tutor' is equivalent to 'dog owner' or 'caregiver.' According to the reviewer's suggestion, we decided to substitute the word 'tutor' with 'caregiver' throughout the manuscript

Line 108 please put a space after <

Corrected

Table 1 – the Mean should be a number < 50 mm and > 50 mm is not a mean. Please provide means with SD for this. 

The table was corrected, and we divided the mass dimensions into < 50 mm and ≥ 50 mm. We added the number of patients that presented masses in each category.

The reviewer is missing a lot more results. Full Signalment of patients is missing.

We decided to add to table 1 the clinical signs presented by the patients, as well as, the type of tumours registered in each group.

To introduce a new technique, imaging of the tumors with pictures of CTs and measurements would be very helpful. Also mass dimensions with just one dimension present is not sufficient in advanced oncology. Furthermore information regarding blood loss would be great to compare the two techniques.

Information about blood loss were referred at discussion section: “The use of a CO2 laser in the nasopharyngeal region was found to be associated with significantly reduce intraoperative bleeding compared to conventional scalpel-based surgery ”

Additionally since the owners are allowed to choose the procedure they like done rather than randomly being assigned to a group more understanding on counseling and how the decisions were derived is important.

We completely agree with the reviewer that it is important to provide references explaining why some patients underwent scalpel blade surgery while others underwent CO2 laser surgery. Therefore, we decided to add the following sentence to the discussion section“After discussing with caregivers about the benefits of the CO2 laser surgery technique, such as reduced bleeding, lower patient anesthesia consumption, shorter surgery time, less post-operative pain, faster healing, and improved local scarring compared to the conventional scalpel blade technique, caregivers were able to choose which technique to use. Those who chose the conventional scalpel blade technique justified their decision only based on the economic difference between the two techniques.”

Please provide the anesthesia and analgesic protocol for the reviewers.

We totally agree with the reviewer that information about anesthesia and analgesic protocol should be included, so we added the following sentence to the Materials & Methods section: “All patients received the same therapeutic protocol with fluid therapy NaCl 0.9% (2 ml/kg/hr), Cefazolin (20 mg/kg), Buprenorphine (10 μg/kg), Acepromazine (0.02 mg/kg), and Carprofen (6 mg/kg). For anesthesia induction, Propofol (4 ml/kg) was used, and maintenance was done with isoflurane with a dose of 1.0%”.

How was healing time evaluated? Were patients reexamined under sedation to examine the incision site? in line 188

We totally agree with the reviewer that information about the time point for healing evaluation should be included. Therefore, we added the following sentence to the Materials & Methods section:“Evaluation of the local scar tissue was performed by the surgeon in all patients at day 10 post-surgery, by opening the patient mouth to visualize the scar's appearance and touching it with the index finger to assess its softness or roughness. Sedation was not necessary for this procedure in any patient.”

Line 180 Lower edema level was assessed how? If patients were not thoroughly examined post surgery this is hard to compare and an explanation needs to be given that is more detailed than this.

Answered in the previous question

Please provide information about follow up treatment for each case. Since there are only 6 cases a table with each case can be provided with vital information to better follow the cases. 

We totally agree with the reviewer about the importance of providing information about the follow-up period. Therefore, we added the following sentence to the Materials & Methods section: “All patients underwent follow-up over a period of 6 months (180 days) during the study.”

Round 2

Reviewer 2 Report

Comments and Suggestions for Authors

After reviewing the paper, the reviewer considers that with the changes made by the authors, the work is suitable for publication.

Author Response

Dear Reviewer,

I hope this message finds you well. Thank you for taking the time to review our manuscript titled "CO2 LASER VERSUS BLADE SCALPEL SURGERY IN THE MANAGEMENT OF NASOPHARYNGEAL MASSES IN DOGS," which we have submitted for publication in the Animals. We appreciate your valuable feedback, and we have carefully considered all of your comments. In response, we have made revisions to the manuscript to address each point raised. Below, you will find our detailed responses to your comments, organized to correspond with the changes made in the original version of the manuscript.

Thank you once again for your time and thoughtful input.

Best regards,

After reviewing the paper, the reviewer considers that with the changes made by the authors, the work is suitable for publication.

We thank the reviewer for their work, assistance and time in improving the original manuscript, making it easier for readers to understand. Thank you.

Reviewer 3 Report

Comments and Suggestions for Authors

Review

CO2 LASER Versus Blade Scalpel Surgery in the Management of Nasopharyngeal Masses in Dogs

The reviewer thanks the authors for this re-submission. The manuscript is much improved and many questions are answered. A few items still remain.

The overall study is set up like a prospective study. It is still not fully clear if this is the case. There is no beginning or end date to the study . Over what time frame was this collected? 5 years 10 years the start and end of the study has to be named for example the study started Jan 1, 2020 and ended Dec 31, 2022. – please address this point.

Line 33 for p-values  < 0.05 not > 0.05.

Line 34 change (p=0.038)   to  (p = 0.038)  The p is in italic and there is are space between. Please change throughout the manuscript!

Line 68 The CO2 laser’s falls  - Please strike the ‘s. It should say: The CO2 laser falls

Line 72 Add space after resections.

Line 85 – change the line to their caregivers signed an informed consent allowing participation in the study.

Line 89 – a three view radiographic view including a right and left lateral and a vd pelvis are the standard for metastatic check. Was a VD pelvis done?

Line 90 Please change to: A head CT was obtained of all patients to assess the extend of the tumor

Line 98  change fluidotherapy to fluid therapy.

Line 98 99 please add the route of administration to all medications used and frequency of administration.

Line 98 99 Carprofen 6mg/kg – that is a high dose. Usually a 4.4 mg/kg once daily oral dose is administered. Is this correct?

Line 98 99 – Please exchange Kg to kg in all medication doses

Line 100 – dose of 1% of isoflurane seems unusual. Usually we adjust isoflurane according to the need for the patient – anesthesia personal may increase to 1-2% or decrease a little to 0.5-1 % depending on what the patient needs. If the authors just say anesthesia was maintained with isoflurane that should be enough.

Line 110 – Please change to To assess the patient’s postoperative pain level, …..

Line 124 – please change to : Twelve patients were enrolled into the study. They were divided equally between the GS …..

Table 1 – Tumor dimensions need to be reported here. We need length, width and height of the tumors here. There are CTs available for all patients so it should be able to report all three dimensions of the tumors

Line 135 strike “will”

Line 143 – Table 1 : There is a discrepancy between the tumor diagnosis of the written information in Line 143-145 and Table 1 See below. Please clarify. This is concerning.

The 141 histopathological diagnosis of the excised nasopharyngeal masses was very similar in 142 both groups, with the following results:

adenocarcinoma (3 in GS and 3 in GL) being the most common, followed by

fibrosarcoma (2 in GS and 1 in GL) and

lymphoma (1 in GS 144 and 1 GL).

Melanoma was only registered in one patient in the GL group.

Table 1

Type of tumor n Scalpel (n) CO2 laser (n)

Squamous cell carcinoma 4 3 1

Sarcoma 3 2 1

Fibrosarcoma 2 1 1

Undifferentiated sarcoma 2 - 2

Lymphoma 1 - 1

Line189 – exchange tutour with owners

Line 272 – change thie to this

Line 288 – please end the statement of the conclusion after this: ….a promising advancement in veterinary surgical practice for treating nasopharyngeal masses. Please strike the rest of the sentence : , offering improvement ….anatomical area. This last part is overreaching.

Comments on the Quality of English Language

English only needs minor changes. 

Author Response

Dear Reviewer,

I hope this message finds you well. Thank you for taking the time to review our manuscript titled "CO2 LASER VERSUS BLADE SCALPEL SURGERY IN THE MANAGEMENT OF NASOPHARYNGEAL MASSES IN DOGS," which we have submitted for publication in the Animals. We appreciate your valuable feedback, and we have carefully considered all of your comments. In response, we have made revisions to the manuscript to address each point raised. Below, you will find our detailed responses to your comments, organized to correspond with the changes made in the original version of the manuscript.

Thank you once again for your time and thoughtful input.

Best regards,

The reviewer thanks the authors for this re-submission. The manuscript is much improved and many questions are answered. A few items still remain.

We thank the reviewer for their work, assistance and time in improving the original manuscript, making it easier for readers to understand. Thank you.

The overall study is set up like a prospective study. It is still not fully clear if this is the case. There is no beginning or end date to the study . Over what time frame was this collected? 5 years 10 years the start and end of the study has to be named for example the study started Jan 1, 2020 and ended Dec 31, 2022. – please address this point.

In accordance with the reviewer's suggestion, we decided to change the sentence  The study involved a sample of 12 dogs (N=12) of both genders, breeds, and ages, all diagnosed with malignant nasopharyngeal processes initially characterized by cytology.” to “The study started on January 1st, 2022, and concluded on January 31st, 2024. It involved a sample of 12 dogs (N=12) of both genders, breeds, and ages, all diagnosed with malignant nasopharyngeal processes initially characterized by cytology.”

Line 33 for p-values  < 0.05 not > 0.05.

Corrected

Line 34 change (p=0.038)   to  (p = 0.038)  The is in italic and there is are space between. Please change throughout the manuscript!

Corrected

Line 68 The COlaser’s falls  - Please strike the ‘s. It should say: The CO2 laser falls

Corrected

Line 72 Add space after resections.

Corrected

Line 85 – change the line to their caregivers signed an informed consent allowing participation in the study.

Corrected

Line 89 – a three view radiographic view including a right and left lateral and a vd pelvis are the standard for metastatic check. Was a VD pelvis done?

No VD pelvis radiographic was made for the patients.

Line 90 Please change to: A head CT was obtained of all patients to assess the extend of the tumor

Corrected

Line 98  change fluidotherapy to fluid therapy.

Corrected

Line 98 99 please add the route of administration to all medications used and frequency of administration.

Corrected. All medications were administered only once. This also included the antibiotic, as the duration of all surgeries was always less than 120 minutes and no significant blood loss were registered, thus not requiring its repetition.

Line 98 99 Carprofen 6mg/kg – that is a high dose. Usually a 4.4 mg/kg once daily oral dose is administered. Is this correct?

The

There was an error in the dosage writing. It was corrected.

Line 98 99 – Please exchange Kg to kg in all medication doses

Corrected

Line 100 – dose of 1% of isoflurane seems unusual. Usually we adjust isoflurane according to the need for the patient – anesthesia personal may increase to 1-2% or decrease a little to 0.5-1 % depending on what the patient needs. If the authors just say anesthesia was maintained with isoflurane that should be enough.

Corrected

Line 110 – Please change to To assess the patient’s postoperative pain level, …..

Corrected

Line 124 – please change to : Twelve patients were enrolled into the study. They were divided equally between the GS …..

Corrected

Table 1 – Tumor dimensions need to be reported here. We need length, width and height of the tumors here. There are CTs available for all patients so it should be able to report all three dimensions of the tumors

We reviewed all the CT scans from the enrolled patients and added the information for their dimensions to Table 1, as per the reviewer's suggestions.

~Line 135 strike “will”

Corrected

Line 143 – Table 1 : There is a discrepancy between the tumor diagnosis of the written information in Line 143-145 and Table 1 See below. Please clarify. This is concerning. 

The sentence was replaced by the folowing: “The histopathological diagnosis of the excised nasopharyngeal masses was similar in both groups, with the following results: squamous cell carcinoma (4 cases in total, 3 in the GS and 1 in the GL) was the most prevalent, followed by adenocarcinoma (3 cases in total, 2 in the GS and 1 in the GL), fibrosarcoma (2 cases in total, 1 in each group), undifferentiated sarcoma (2 cases in total, only in the GL), and lymphoma (1 case in total, only in the GL). Additionally, a melanoma was concurrently detected in one patient in the GL, although it was not associated with the nasopharynx but rather with the lower lip of the patient.“

Line189 – exchange tutour with owners

Corrected

Line 272 – change thie to this

Corrected

Line 288 – please end the statement of the conclusion after this: ….a promising advancement in veterinary surgical practice for treating nasopharyngeal masses. Please strike the rest of the sentence : , offering improvement ….anatomical area. This last part is overreaching.

Corrected